# ON THE TRANSFER OF DISENTANGLED REPRESENTATIONS IN REALISTIC SETTINGS

**Andrea Dittadi,**[*†1] **Frederik Träuble,**[*2] **Francesco Locatello,**[2,3] **Manuel Wüthrich,**[2]
**Vaibhav Agrawal,**[2] **Ole Winther,**[1,4,5] **Stefan Bauer,**[2,6] **Bernhard Schölkopf**[2]

[1]Technical University of Denmark, [2]Max Planck Institute for Intelligent Systems,
[3]ETH Zurich, Department for Computer Science, [4]Copenhagen University Hospital,
[5]University of Copenhagen , [6]CIFAR Azrieli Global Scholar

## ABSTRACT

Learning meaningful representations that disentangle the underlying structure of the data generating process is considered to be of key importance in machine learning. While disentangled representations were found to be useful for diverse tasks such as abstract reasoning and fair classification, their scalability and real-world impact remain questionable. We introduce a new high-resolution dataset with 1M simulated images and over 1,800 annotated real-world images of the same setup. In contrast to previous work, this new dataset exhibits correlations, a complex underlying structure, and allows to evaluate transfer to unseen simulated and real-world settings where the encoder i) remains in distribution or ii) is out of distribution. We propose new architectures in order to scale disentangled representation learning to realistic high-resolution settings and conduct a large-scale empirical study of disentangled representations on this dataset. We observe that disentanglement is a good predictor for out-of-distribution (OOD) task performance.

## 1 INTRODUCTION

Disentangled representations hold the promise of generalization to unseen scenarios (Higgins et al., 2017b), increased interpretability (Adel et al., 2018; Higgins et al., 2018) and faster learning on downstream tasks (van Steenkiste et al., 2019; Locatello et al., 2019a). However, most of the focus in learning disentangled representations has been on small synthetic datasets whose ground truth factors exhibit perfect independence by design. More realistic settings remain largely unexplored. We hypothesize that this is because real-world scenarios present several challenges that have not been extensively studied to date. Important challenges are scaling (much higher resolution in observations and factors), occlusions, and

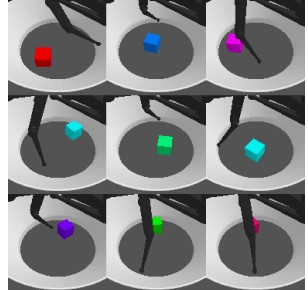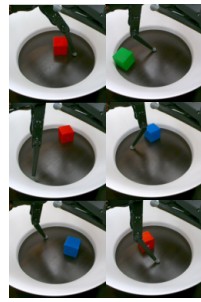

Figure 1: Images from the simulated dataset (left) and from the real-world setup (right).

correlation between factors. Consider, for instance, a robotic arm moving a cube: Here, the robot arm can occlude parts of the cube, and its end-effector position exhibits correlations with the cube's position and orientation, which might be problematic for common disentanglement learners (Träuble et al., 2020). Another difficulty is that we typically have only limited access to ground truth labels in the real world, which requires robust frameworks for model selection when no or only weak labels are available.

---

*Equal contribution. Correspondence to: <adit@dtu.dk>, <frederik.traeuble@tuebingen.mpg.de>.
†Work done during an internship at the Max Planck Institute for Intelligent Systems.

The goal of this work is to provide a path towards disentangled representation learning in realistic settings. First, we argue that this requires a new dataset that captures the challenges mentioned above. We propose a dataset consisting of simulated observations from a scene where a robotic arm interacts with a cube in a stage (see Fig. 1). This setting exhibits correlations and occlusions that are typical in real-world robotics. Second, we show how to scale the architecture of disentanglement methods to perform well on this dataset. Third, we extensively analyze the usefulness of disentangled representations in terms of out-of-distribution downstream generalization, both in terms of held-out factors of variation and sim2real transfer. In fact, our dataset is based on the TriFinger robot from Wüthrich et al. (2020), which can be built to test the deployment of models in the real world. While the analysis in this paper focuses on the transfer and generalization of predictive models, we hope that our dataset may serve as a benchmark to explore the usefulness of disentangled representations in real-world control tasks.

The contributions of this paper can be summarized as follows:

- We propose a new dataset for disentangled representation learning, containing 1M simulated high-resolution images from a robotic setup, with seven partly correlated factors of variation. Additionally, we provide a dataset of over 1,800 annotated images from the corresponding real-world setup that can be used for challenging sim2real transfer tasks. These datasets are made publicly available.[1]
- We propose a new neural architecture to successfully scale VAE-based disentanglement learning approaches to complex datasets.
- We conduct a large-scale empirical study on generalization to various transfer scenarios on this challenging dataset. We train 1,080 models using state-of-the-art disentanglement methods and discover that disentanglement is a good predictor for out-of-distribution (OOD) performance of downstream tasks.

## 2 RELATED WORK

**Disentanglement methods.** Most state-of-the-art disentangled representation learning approaches are based on the framework of variational autoencoders (VAEs) (Kingma & Welling, 2014; Rezende et al., 2014). A (high-dimensional) observation $\boldsymbol{x}$ is assumed to be generated according to the latent variable model $p_\theta(\boldsymbol{x}|\boldsymbol{z})p(\boldsymbol{z})$ where the latent variables $\boldsymbol{z}$ have a fixed prior $p(\boldsymbol{z})$. The generative model $p_\theta(\boldsymbol{x}|\boldsymbol{z})$ and the approximate posterior distribution $q_\phi(\boldsymbol{z}|\boldsymbol{x})$ are typically parameterized by neural networks, which are optimized by maximizing the evidence lower bound (ELBO):

$$\mathcal{L}_{VAE} = \mathbb{E}_{q_\phi(\boldsymbol{z}|\boldsymbol{x})}[\log p_\theta(\boldsymbol{x}|\boldsymbol{z})] - D_{\mathrm{KL}}(q_\phi(\boldsymbol{z}|\boldsymbol{x})\|p(\boldsymbol{z})) \leq \log p(\boldsymbol{x}) \tag{1}$$

As the above objective does not enforce any structure on the latent space except for some similarity to $p(\boldsymbol{z})$, different regularization strategies have been proposed, along with evaluation metrics to gauge the disentanglement of the learned representations (Higgins et al., 2017a; Kim & Mnih, 2018; Burgess et al., 2018; Kumar et al., 2018; Chen et al., 2018; Eastwood & Williams, 2018). Recently, Locatello et al. (2019b, Theorem 1) showed that the purely unsupervised learning of disentangled representations is impossible. This limitation can be overcome without the need for explicitly labeled data by introducing weak labels (Locatello et al., 2020; Shu et al., 2019). Ideas related to disentangling the factors of variation date back to the non-linear ICA literature (Comon, 1994; Hyvärinen & Pajunen, 1999; Bach & Jordan, 2002; Jutten & Karhunen, 2003; Hyvarinen & Morioka, 2016; Hyvarinen et al., 2019; Gresele et al., 2019). Recent work combines non-linear ICA with disentanglement (Khemakhem et al., 2020; Sorrenson et al., 2020; Klindt et al., 2020).

**Evaluating disentangled representations.** The *BetaVAE* (Higgins et al., 2017a) and *Factor-VAE* (Kim & Mnih, 2018) scores measure disentanglement by performing an intervention on the factors of variation and predicting which factor was intervened on. The *Mutual Information Gap (MIG)* (Chen et al., 2018), *Modularity* (Ridgeway & Mozer, 2018), *DCI Disentanglement* (Eastwood & Williams, 2018) and *SAP* scores (Kumar et al., 2018) are based on matrices relating factors of variation and codes (e.g. pairwise mutual information, feature importance and predictability).

---

[1] http://people.tuebingen.mpg.de/ei-datasets/iclr_transfer_paper/robot_finger_datasets.tar (6.18 GB)

**Datasets for disentanglement learning.** *dSprites* (Higgins et al., 2017a), which consists of binary low-resolution 2D images of basic shapes, is one of the most commonly used synthetic datasets for disentanglement learning. *Color-dSprites*, *Noisy-dSprites*, and *Scream-dSprites* are slightly more challenging variants of dSprites. The *SmallNORB* dataset contains toy images rendered under different lighting conditions, elevations and azimuths (LeCun et al., 2004). *Cars3D* (Reed et al., 2015) exhibits different car models from Fidler et al. (2012) under different camera viewpoints. *3dshapes* is a popular dataset of simple shapes in a 3D scene (Kim & Mnih, 2018). Finally, Gondal et al. (2019) proposed *MPI3D*, containing images of physical 3D objects with seven factors of variation, such as object color, shape, size and position available in a simulated, simulated and highly realistic rendered simulated variant. Except MPI3D which has over 1M images, the size of the other datasets is limited with only $17,568$ to $737,280$ images. All of the above datasets exhibit perfect independence of all factors, the number of possible states is on the order of 1M or less, and due to their static setting they do not allow for dynamic downstream tasks such as reinforcement learning. In addition, except for SmallNORB, the image resolution is limited to 64x64 and there are no occlusions.

**Other related work.** Locatello et al. (2020) probed the out-of-distribution generalization of downstream tasks trained on disentangled representations. However, these representations are trained on the entire dataset. Generalization and transfer performance especially for representation learning has likewise been studied in Dayan (1993); Muandet et al. (2013); Heinze-Deml & Meinshausen (2017); Rojas-Carulla et al. (2018); Suter et al. (2019); Li et al. (2018); Arjovsky et al. (2019); Krueger et al. (2020); Gowal et al. (2020). For the role of disentanglement in causal representation learning we refer to the recent overview by Schölkopf et al. (2021). Träuble et al. (2020) systematically investigated the effects of correlations between factors of variation on disentangled representation learners. Transfer of learned disentangled representations from simulation to the real world has been recently investigated by Gondal et al. (2019) on the MPI3D dataset, and previously by Higgins et al. (2017b) in the context of reinforcement learning. Sim2real transfer is of major interest in the robotic learning community, because of limited data and supervision in the real world (Tobin et al., 2017; Rusu et al., 2017; Peng et al., 2018; James et al., 2019; Yan et al., 2020; Andrychowicz et al., 2020).

## 3 SCALING DISENTANGLED REPRESENTATIONS TO COMPLEX SCENARIOS

**A new challenging dataset.** Simulated images in our dataset are derived from the trifinger robot platform introduced by Wüthrich et al. (2020). The motivation for choosing this setting is that (1) it is challenging due to occlusions, correlations, and other difficulties encountered in robotic settings, (2) it requires modeling of fine details such as tip links at high resolutions, and (3) it corresponds to a robotic setup, so that learned representations can be used for control and reinforcement learning in simulation and in the real world. The scene comprises a robot finger with three joints that can be controlled to manipulate a cube in a bowl-shaped stage. Fig. 1 shows examples of scenes from our dataset. The data is generated from 7 different factors of variation (FoV)

| FoV | Values |
| --- | --- |
| Upper joint | 30 values in $[-0.65, +0.65]$ |
| Middle joint | 30 values in $[-0.5, +0.5]$ |
| Lower joint | 30 values in $[-0.8, +0.8]$ |
| Cube position x | 30 values in $[-0.11, +0.11]$ |
| Cube position y | 30 values in $[-0.11, +0.11]$ |
| Cube rotation | 10 values in $[0°, 81°]$ |
| Cube color hue | 12 values in $[0°, 330°]$ |

Table 1: Factors of variation in the proposed dataset. Values are linearly spaced in the specified intervals. Joint angles are in radians, cube positions in meters.

listed in Table 1. Unlike in previous datasets, not all FoVs are independent: The end-effector (the tip of the finger) can collide with the floor or the cube, resulting in infeasible combinations of the factors (see Appendix B.1). We argue that such correlations are a key feature in real-world data that is not present in existing datasets. The high FoV resolution results in approximately 1.52 billion feasible states, but the dataset itself only contains one million of them (approximately 0.065% of all possible FoV combinations), realistically rendered into $128 \times 128$ images. Additionally, we recorded an annotated dataset under the same conditions in the real-world setup: we acquired 1,809 camera images from the same viewpoint and recorded the labels of the 7 underlying factors of variation. This dataset can be used for out-of-distribution evaluations, few-shot learning, and testing other sim2real aspects.

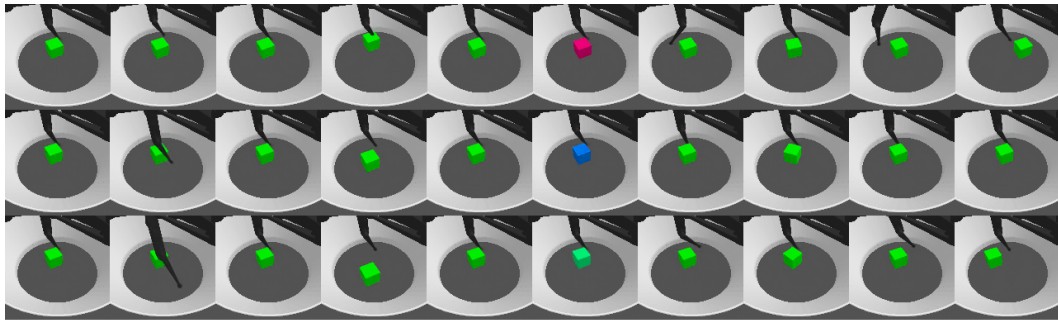

Figure 2: Latent traversals of a trained model that perfectly disentangles the dataset's FoVs. In each column, all latent variables but one are fixed.

**Model architecture.** When scaling disentangled representation learning to more complex datasets, such as the one proposed here, one of the main bottlenecks in current VAE-based approaches is the flexibility of the encoder and decoder networks. In particular, using the architecture from Locatello et al. (2019b), none of the models we trained correctly captured all factors of variation or yielded high-quality reconstructions. While the increased image resolution already presents a challenge, the main practical issue in our new dataset is the level of detail that needs to be modeled. In particular, we identified the cube rotation and the lower joint position to be the factors of variation that were the hardest to capture. This is likely because these factors only produce relatively small changes in the image and hence the reconstruction error.

To overcome these issues, we propose a deeper and wider neural architecture than those commonly used in the disentangled representation learning literature, where the encoder and decoder typically have 4 convolutional and 2 fully-connected layers. Our encoder consists of a convolutional layer, 10 residual blocks, and 2 fully-connected layers. Some residual blocks are followed by 1x1 convolutions that change the number of channels, or by average pooling that downsamples the tensors by a factor of 2 along the spatial dimensions. Each residual block consists of two 3x3 convolutions with a leaky ReLU nonlinearity, and a learnable scalar gating mechanism (Bachlechner et al., 2020). Overall, the encoder has 23 convolutional layers and 2 fully connected layers. The decoder mirrors this architecture, with average pooling replaced by bilinear interpolation for upsampling. The total number of parameters is approximately 16.3M. See Appendix A for further implementation details.

**Experimental setup.** We perform a large-scale empirical study on the simulated dataset introduced above by training 1,080 $\beta$-VAE models.[2] For further experimental details we refer the reader to Appendix A. The hyperparameter sweep is defined as follows:

- We train the models using either unsupervised learning or weakly supervised learning (Locatello et al., 2020). In the weakly supervised case, a model is trained with pairs of images that differ in $k$ factors of variation. Here we fix $k = 1$ as it was shown to lead to higher disentanglement by Locatello et al. (2020). The dataset therefore consists of 500k pairs of images that differ in only one FoV.

- We vary the parameter $\beta$ in $\{1, 2, 4\}$, and use linear deterministic warm-up (Bowman et al., 2015; Sønderby et al., 2016) over the first $\{0, 10000, 50000\}$ training steps.

- The latent space dimensionality is in $\{10, 25, 50\}$.

- Half of the models are trained with additive noise in the input image. This choice is motivated by the fact that adding noise to the input of neural networks has been shown to be beneficial for out-of-distribution generalization (Sietsma & Dow, 1991; Bishop, 1995).

- Each of the 108 resulting configurations is trained with 10 random seeds.

**Can we scale up disentanglement learning?** Most of the trained VAEs in our empirical study fully capture all the elements of a scene, correctly model heavy occlusions, and generate detailed,

---

[2]Training these models requires approximately 2.8 GPU years on NVIDIA Tesla V100 PCIe.

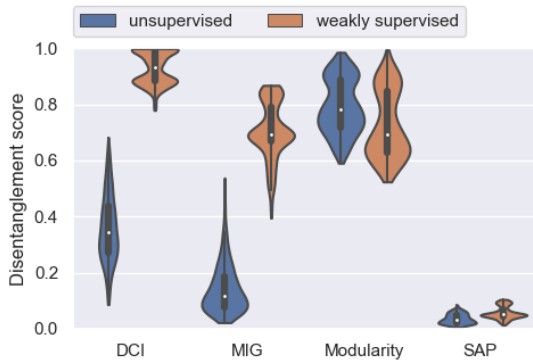 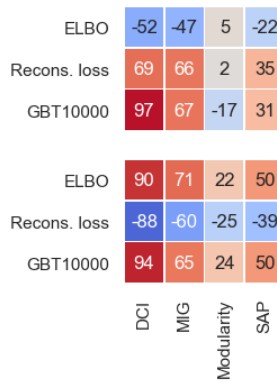

Figure 3: **Left**: Disentanglement metrics aggregating all hyperparameters except for supervision type. **Right**: Rank correlations (Spearman) of ELBO, reconstruction loss, and the test error of a GBT classifier trained on 10,000 labelled data points with disentanglement metrics. The upper rank correlations correspond to the unsupervised models and the lower to the weakly supervised models.

high-quality samples and reconstructions (see Appendix B.2). From visual inspections such as the latent traversals in Fig. 2, we observe that many trained models fully disentangle the ground-truth factors of variation. This, however, appears to only be possible in the weakly supervised scenario. The fact that models trained without supervision learn entangled representations is in line with the impossibility result for the unsupervised learning of disentangled representations from Locatello et al. (2019b). Latent traversals from a selection of models with different degrees of disentanglement are presented in Appendix B.3. Interestingly, the high-disentanglement models seem to correct for correlations and interpolate infeasible states, i.e. the fingertip traverses through the cube or the floor.

**Summary:** The proposed architecture can scale disentanglement learning to more realistic settings, but a form of weak supervision is necessary to achieve high disentanglement.

**How useful are common disentanglement metrics in realistic scenarios?** The violin plot in Fig. 3 (left) shows that DCI and MIG measure high disentanglement under weak supervision and lower disentanglement in the unsupervised setting. This is consistent with our qualitative conclusion from visual inspection of the models (Appendix B.3) and with the aforementioned impossibility result. Many of the models trained with weak supervision exhibit a very high DCI score (29% of them have >99% DCI, some of them up to 99.89%). SAP and Modularity appear to be ineffective at capturing disentanglement in this setting, as also observed by Locatello et al. (2019b). Finally, note that the BetaVAE and FactorVAE metrics are not straightforward to be evaluated on datasets that do not contain all possible combinations of factor values. According to Fig. 3 (right), DCI and MIG strongly correlate with test accuracy of GBT classifiers predicting the FoVs. In the weakly supervised setting, these metrics are strongly correlated with the ELBO (positively) and with the reconstruction loss (negatively). We illustrate these relationships in more detail in Appendix B.4. Such correlations were also observed by Locatello et al. (2020) on significantly less complex datasets, and can be exploited for unsupervised model selection: these unsupervised metrics can be used as proxies for disentanglement metrics, which would require fully labeled data.

**Summary:** DCI and MIG appear to be useful disentanglement metrics in realistic scenarios, whereas other metrics seem to fall short of capturing disentanglement or can be difficult to compute. When using weak supervision, we can select disentangled models with unsupervised metrics.

## 4 FRAMEWORK FOR THE EVALUATION OF OOD GENERALIZATION

Previous work has focused on evaluating the usefulness of disentangled representations for various downstream tasks, such as predicting ground truth factors of variation, fair classification, and abstract reasoning. Here we propose a new framework for evaluating the out-of-distribution (OOD) generalization properties of representations. More specifically, we consider a downstream task – in

our case, regression of ground truth factors – trained on a learned representation of the data, and evaluate the performance on a held-out test set. While the test set typically follows the same distribution as the training set (in-distribution generalization), we also consider test sets that follow a different distribution (out-of-distribution generalization). Our goal is to investigate to what extent, if at all, downstream tasks trained on disentangled representations exhibit a higher degree of OOD generalization than those trained on entangled representations.

Let $D$ denote the training set for disentangled representation learning. To investigate OOD generalization, we train downstream regression models on a subset $D_1 \subset D$ to predict ground truth factor values from the learned representation computed by the encoder. We independently train one predictor per factor. We then test the regression models on a set $D_2$ that differs distributionally from the training set $D_1$, as it either contains images corresponding to held-out values of a chosen FoV (e.g. unseen object colors), or it consists of real-world images. We now differentiate between two scenarios: (1) $D_2 \subset D$, i.e. the OOD test set is a subset of the dataset for representation learning; (2) $D$ and $D_2$ are disjoint and distributionally different. These two scenarios will be denoted by *OOD1* and *OOD2*, respectively. For example, consider the case in which distributional shifts are based on one FoV: the color of the object. Then, we could define these datasets such that images in $D$ always contain a red or blue object, and those in $D_1 \subset D$ always contain a red object. In the OOD1 scenario, images in $D_2$ would always contain a blue object, whereas in the OOD2 case they would always contain an object that is neither red nor blue.

The regression models considered here are Gradient Boosted Trees (GBT), random forests, and MLPs with $\{1, 2, 3\}$ hidden layers. Since random forests exhibit a similar behavior to GBTs, and all MLPs yield similar results to each other, we choose GBTs and the 2-layer MLP as representative models and only report results for those. To quantify prediction quality, we normalize the ground truth factor values to the range $[0, 1]$, and compute the mean absolute error (MAE). Since the values are normalized, we can define our transfer metric as the average of the MAE over all factors (except for the FoV that is OOD).

## 5   Benefits and Transfer of Structured Representations

**Experimental setup.** We evaluate the transfer metric introduced in Section 4 across all 1,080 trained models. To compute this metric, we train regression models to predict the ground truth factors of variation, and test them under distributional shift. We consider distributional shifts in terms of cube color or sim2real, and we do not evaluate downstream prediction of cube color. We report scores for two different regression models: a Gradient Boosted Tree (GBT) and an MLP with 2 hidden layers of size 256. In Appendix A we provide details on the datasets used in this section.

In the OOD1 setting, we have $D_2 \subset D$, hence the encoder is in-distribution: we are testing the predictor on representations of images that were in the training set of the representation learning algorithm. Therefore, we expect the representations to be meaningful. We consider three scenarios:

- OOD1-A: The regression models are trained on 1 cube color (red) and evaluated on the remaining 7 colors.

- OOD1-B: The regression models are trained on 4 cube colors with high hue in the HSV space, and evaluated on 4 cube colors with low hue (extrapolation).

- OOD1-C: The regression models are again trained and evaluated on 4 cube colors, but the training and evaluation colors are alternating along the hue dimension (interpolation).

In the more challenging setting where even the encoder goes out-of-distribution (OOD2, with $D_2 \cap D = \varnothing$), we train the regression models on a subset of the training set $D$ that includes all 8 cube colors, and we consider the two following scenarios:

- OOD2-A: The regression models are evaluated on simulated data, on 4 cube colors that are out of the encoder's training distribution.

- OOD2-B: The regression models are evaluated on real-world images of the robotic setup, without any adaptation or fine-tuning.

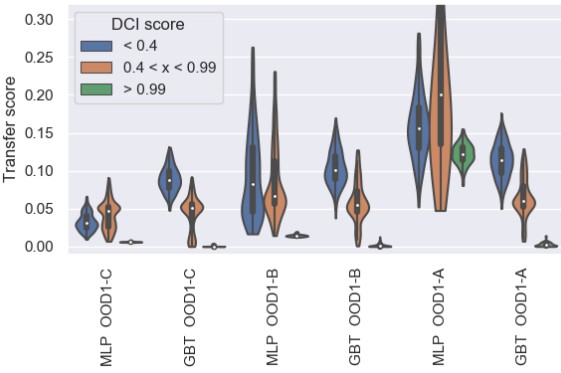 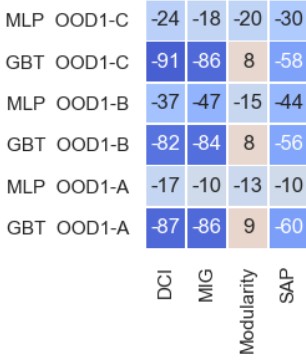

Figure 4: Higher disentanglement corresponds to better generalization across all OOD1 scenarios, as seen from the transfer scores (left). The transfer score is computed as the mean absolute prediction error of ground truth factor values (lower is better). This correlation is particularly evident in the GBT case, whereas MLPs appear to exhibit better OOD1 transfer with very high disentanglement only. These results are mirrored in the Spearman rank correlations between transfer scores and disentanglement metrics (right).

**Is disentanglement correlated with OOD1 generalization?** In Fig. 4 we consistently observe a negative correlation between disentanglement and transfer error across all OOD1 settings. The correlation is mild when using MLPs, strong when using GBTs. This difference is expected, as GBTs have an axis-alignment bias whereas MLPs can – given enough data and capacity – disentangle an entangled representation more easily. Our results therefore suggest that **highly disentangled representations are useful for generalizing out-of-distribution as long as the encoder remains in-distribution**. This is in line with the correlation found by Locatello et al. (2019b) between disentanglement and the GBT10000 metric. There, however, GBTs are tested on the same distribution as the training distribution, while here we test them under distributional shift. Given that the computation of disentanglement scores requires labels, this is of little benefit in the unsupervised setting. However, it can be exploited in the weakly supervised setting, where disentanglement was shown to correlate with ELBO and reconstruction loss (Section 3). Therefore, model selection for representations that transfer well in these scenarios is feasible based on the ELBO or reconstruction loss, when weak supervision is available. Note that, in absolute terms, the OOD generalization error with encoder in-distribution (OOD1) is very low in the high-disentanglement case (the only exception being the MLP in the OOD1-C case, with the 1-7 color split, which seems to overfit). This suggests that disentangled representations can be useful in downstream tasks even when transferring out of the training distribution.

**Summary:** Disentanglement seems to be positively correlated with OOD generalization of downstream tasks, provided that the encoder remains in-distribution (OOD1). Since in the weakly supervised case disentanglement correlates with the ELBO and the reconstruction loss, model selection can be performed using these metrics as proxies for disentanglement. These metrics have the advantage that they can be computed without labels, unlike disentanglement metrics.

**Is disentanglement correlated with OOD2 generalization?** As seen in Fig. 5, the negative correlation between disentanglement and GBT transfer error is weaker when the encoder is out of distribution (OOD2). Nonetheless, we observe a non-negligible correlation for GBTs in the OOD2-A case, where we investigate out-of-distribution generalization along one FoV, with observations in $D_2$ still generated from the same simulator. In the OOD2-B setting, where the observations are taken from cameras in the corresponding real-world setting, the correlation between disentanglement and transfer performance appears to be minor at best. This scenario can be considered a variant of zero-shot sim2real generalization.

**Summary:** Disentanglement has a minor effect on out-of-distribution generalization outside of the training distribution of the encoder (OOD2).

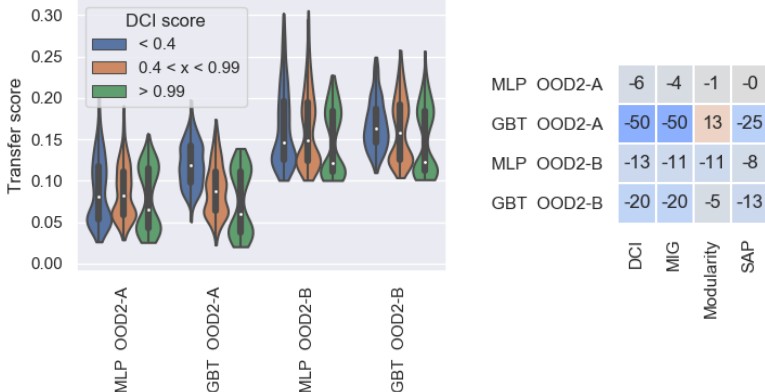

Figure 5: Disentanglement affects generalization across the OOD2 scenarios only minimally as seen from transfer scores (left) and corresponding rank correlations with disentanglement metrics (right).

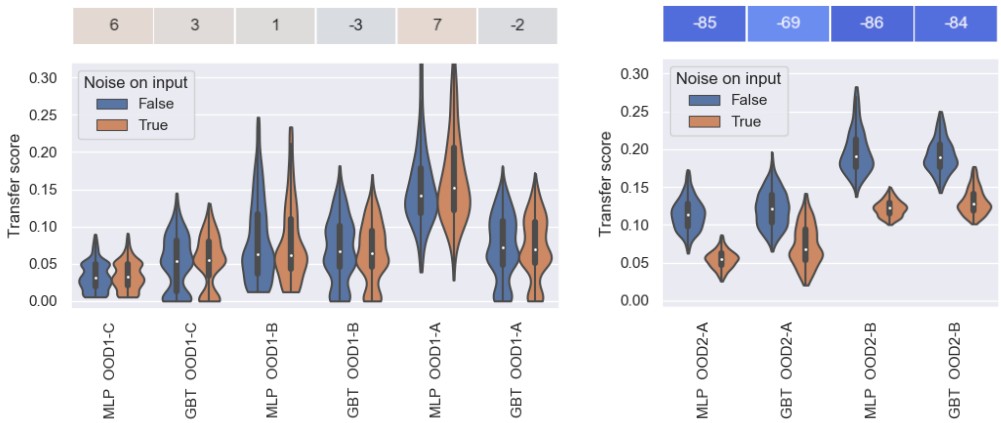

Figure 6: Noise improves generalization across the OOD2 scenarios and less so for the OOD1 scenarios as seen from the transfer scores. Top row: Spearman rank correlation coefficients between transfer metrics and presence of noise in the input.

**What else matters for OOD2 generalization?** Results in Fig. 6 suggest that adding Gaussian noise to the input during training as described in Section 3 leads to significantly better OOD2 generalization, and has no effect on OOD1 generalization. Adding noise to the input of neural networks is known to lead to better generalization (Sietsma & Dow, 1991; Bishop, 1995). This is in agreement with our results, since OOD1 generalization does not require generalization of the encoder, while OOD2 does. Interestingly, closer inspection reveals that the contribution of different factors of variation to the generalization error can vary widely. See Appendix B.5 for further details. In particular, with noisy input, the position of the cube is predicted accurately even in real-world images ($<5\%$ mean absolute error on each axis). This is promising for robotics applications, where the true state of the joints is observable but inference of the cube position relies on object tracking methods. Fig. 7 shows an example of real-world inputs and reconstructions of their simulated equivalents.

**Summary:** Adding input noise during training appears to be significantly beneficial for OOD2 generalization, while having no effect when the encoder is kept in its training distribution (OOD1).

## 6    CONCLUSION

Despite the growing importance of the field and the potential societal impact in the medical domain (Chartsias et al., 2018) and fair decision making (Locatello et al., 2019a), state-of-the-art approaches for learning disentangled representations have so far only been systematically evaluated on synthetic toy datasets. Here we introduced a new high-resolution dataset with 1M simulated images and over 1,800 annotated real-world images of the same setup. This dataset

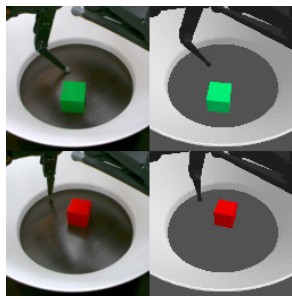

exhibits a number of challenges and features which are not present in previous datasets: it contains correlations between factors, occlusions, a complex underlying structure, and it allows for evaluation of transfer to unseen simulated and real-world settings. We proposed a new VAE architecture to scale disentangled representation learning to this realistic setting and conducted a large-scale empirical study of disentangled representations on this dataset. We discovered that disentanglement is a good predictor of OOD generalization of downstream tasks and showed that, in the context of weak supervision, model selection for good OOD performance can be based on the ELBO or the reconstruction loss, which are accessible without explicit labels. Our setting allows for studying a wide variety of interesting downstream tasks in the future, such as reinforcement learning or learning a dynamics model of the environment. Finally, we believe that in the future it will be important to take further steps in the direction of this paper by considering settings with even more complex structures and stronger correlations between factors.

Figure 7: Zero-shot transfer to real-world observations of our models trained in simulation. Left: input; right: reconstruction.

ACKNOWLEDGEMENTS

The authors thank Shruti Joshi and Felix Widmaier for their useful comments on the simulated setup, Anirudh Goyal for helpful discussions and comments, and CIFAR for the support. We thank the International Max Planck Research School for Intelligent Systems (IMPRS-IS) for supporting Frederik Träuble.

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

## A   IMPLEMENTATION DETAILS

**Training.**   We train the $\beta$-VAEs by maximizing the following objective function:

$$\mathcal{L}_{VAE}^{\beta} = \mathbb{E}_{q_\phi(\boldsymbol{z}|\boldsymbol{x})}[\log p_\theta(\boldsymbol{x}|\boldsymbol{z})] - \beta D_{\mathrm{KL}}(q_\phi(\boldsymbol{z}|\boldsymbol{x})\|p(\boldsymbol{z})) \leq \log p(\boldsymbol{x})$$

with $\beta > 0$ using the Adam optimizer (Kingma & Ba, 2014) with default parameters. We use a batch size of 64 and train for 400k steps. The learning rate is initialized to 1e-4 and halved at 150k and 300k training steps. We clip the global gradient norm to 1.0 before each weight update. Following Locatello et al. (2019b), we use a Gaussian encoder with an isotropic Gaussian prior for the latent variable, and a Bernoulli decoder. Our implementation of weakly supervised learning is based on Ada-GVAE (Locatello et al., 2020), but uses a symmetrized KL divergence:

$$\tilde{D}_{\mathrm{KL}}(p, q) = \frac{1}{2} D_{\mathrm{KL}}(p\|q) + \frac{1}{2} D_{\mathrm{KL}}(q\|p)$$

to infer which latent dimensions should be aggregated.

The noise added to the encoder's input consists of two independent components, both iid Gaussian with zero mean: one is independent for each subpixel (RGB) and has standard deviation 0.03, the other is a $8 \times 8$ pixel-wise (greyscale) noise with standard deviation 0.15, bilinearly upsampled by a factor of 16. The latter has been designed (by visual inspection) to roughly mimic observation noise in the real images due to complex lighting conditions.

**Neural architecture.**   Architectural details are provided in Tables 2 and 3, and Fig. 8 provides a high-level overview. In preliminary experiments, we observed that batch normalization, layer normalization, and dropout did not significantly affect performance in terms of ELBO, model samples, and disentanglement scores, both in the unsupervised and weakly supervised settings. On the other hand, layer normalization before the posterior parameterization (last layer of the encoder) appeared to be beneficial for stability in early training. While using an architecture based on residual blocks leads to fast convergence, in practice we observed that it may be challenging to keep the gradients in check at the beginning of training.[3] In order to solve this issue, we resorted to a simple scalar gating mechanism in the residual blocks (Bachlechner et al., 2020) such that each residual block is initialized to the identity.

**Datasets and OOD evaluation.**   Because we evaluate OOD generalization in terms of cube color hue (except in the sim2real case), we first sampled 8 color hues at random from the 12 specified in Table 1. The chosen hues are: $[0°, 120°, 150°, 180°, 210°, 270°, 300°, 330°]$. Then, the dataset $D$ used for training VAEs is generated by randomly sampling values for the factors of variation from Table 1, with the color hue restricted to the above-mentioned values. This makes OOD2 evaluation possible, specifically OOD2-A where the learned predictors are tested on representations extracted from images with held-out values of the cube hue.

For evaluation of out-of-distribution generalization, we train the downstream predictors on a subset $D_1 \subset D$ of the representation training set. The downstream training set $D_1$ is sampled at random from $D$ but only contains a (not necessarily proper) subset of the 8 cube colors. This subset contains 1 color in the OOD1-A case, 4 colors in OOD1-B and OOD1-C, all 8 colors in OOD2 (in this case $D_1$ is simply a random subset of $D$). Then we test the downstream predictors on a set $D_2$ distributionally different from $D_1$ in terms of cube color (all OOD1 scenarios as well as OOD2-A) or sim2real (OOD2-B). In the OOD1 case, $D_2$ is also a subset of $D$ and is generated the same way. In each OOD1 case, the test set $D_2$ is paired with its corresponding $D_1$ that was used to train the downstream predictors. $D_2$ contains all colors in $D$ minus those in $D_1$. In the OOD2-A case, $D_2$ is a separate dataset containing 5k simulated images like those in $D$, except that these only contain the 4 colors that were left out from the VAE training set $D$ (hue in $[30°, 60°, 90°, 240°]$). In the OOD2-B case, the set $D_2$ is the dataset of real images. Following previous work (e.g. the GBT10000 metric in Locatello et al. (2019b)), the training set $D_1$ and test set $D_2$ for downstream tasks contain 10k and 5k images, respectively, except in the OOD2-B case, where the size is limited by the size of the real dataset.

---

[3]This instability may also be exacerbated in probabilistic models by the sampling step in latent space, where a large log variance causes the decoder input to take very large values. Intuitively, this might be a reason why layer normalization before latent space appears to be beneficial for training stability.

| Encoder | | | Decoder | |
| --- | --- | --- | --- | --- |
| **Operation** | **Output Shape** | | **Operation** | **Output Shape** |
| Input | $128 \times 128 \times K$ | | Input | $d$ |
| Conv 5x5, stride 2, 64 ch. | $64 \times 64 \times 64$ | | FC(512) | 512 |
| LeakyReLU(0.02) | — | | LeakyReLU(0.02) | — |
| 2x ResidualBlock(64) | — | | FC(4096) | 4096 |
| Conv 1x1, 128 channels | $64 \times 64 \times 128$ | | Reshape | $4 \times 4 \times 256$ |
| AveragePool(2) | $32 \times 32 \times 128$ | | 2x ResidualBlock(256) | — |
| 2x ResidualBlock(128) | — | | BilinearInterpolation(2) | $8 \times 8 \times 256$ |
| AveragePool(2) | $16 \times 16 \times 128$ | | 2x ResidualBlock(256) | — |
| 2x ResidualBlock(128) | — | | Conv 1x1, 128 channels | $8 \times 8 \times 128$ |
| Conv 1x1, 256 channels | $16 \times 16 \times 256$ | | BilinearInterpolation(2) | $16 \times 16 \times 128$ |
| AveragePool(2) | $8 \times 8 \times 256$ | | 2x ResidualBlock(128) | — |
| 2x ResidualBlock(256) | — | | BilinearInterpolation(2) | $32 \times 32 \times 128$ |
| AveragePool(2) | $4 \times 4 \times 256$ | | 2x ResidualBlock(128) | — |
| 2x ResidualBlock(256) | — | | Conv 1x1, 64 channels | $32 \times 32 \times 64$ |
| Flatten | 4096 | | BilinearInterpolation(2) | $64 \times 64 \times 64$ |
| LeakyReLU(0.02) | — | | 2x ResidualBlock(64) | — |
| FC(512) | 512 | | BilinearInterpolation(2) | $128 \times 128 \times 64$ |
| LeakyReLU(0.02) | — | | LeakyReLU(0.02) | — |
| LayerNorm | — | | Conv 5x5, $K$ channels | $128 \times 128 \times K$ |
| 2x FC($d$) | $2d$ | | | |

Table 2: Encoder (left) and decoder (right) architectures. The latent space dimensionality is denoted by $d$, and $K = 3$ indicates the number of image channels. Last line in the encoder architecture: the fully connected layer parameterizing the log variance of the approximate posterior distributions of the latent variables has custom initialization. The weights are initialized with $1/10$ standard deviation than the default value, and the biases are initialized to $-1$ instead of $0$. Empirically, this together with (learnable) LayerNorm was beneficial for training stability at the beginning of training.

| **Residual Block** |
| --- |
| Input: shape $H \times W \times C$ |
| LeakyReLU(0.02) |
| Conv 3x3, $C$ channels |
| LeakyReLU(0.02) |
| Conv 3x3, $C$ channels |
| Scalar gate |
| Sum with input |

Table 3: Architecture of one residual block. The scalar gate is implemented by multiplying the tensor by a learnable scalar parameter before adding it to the block input. Initializing the residual block to the identity by setting this parameter to zero has been originally proposed by Bachlechner et al. (2020). The tensor shape is constant throughout the residual block.

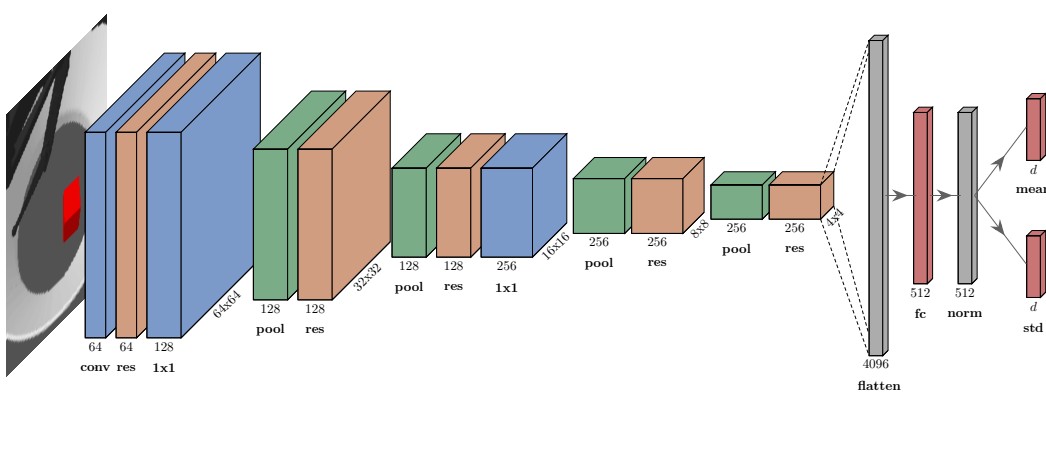

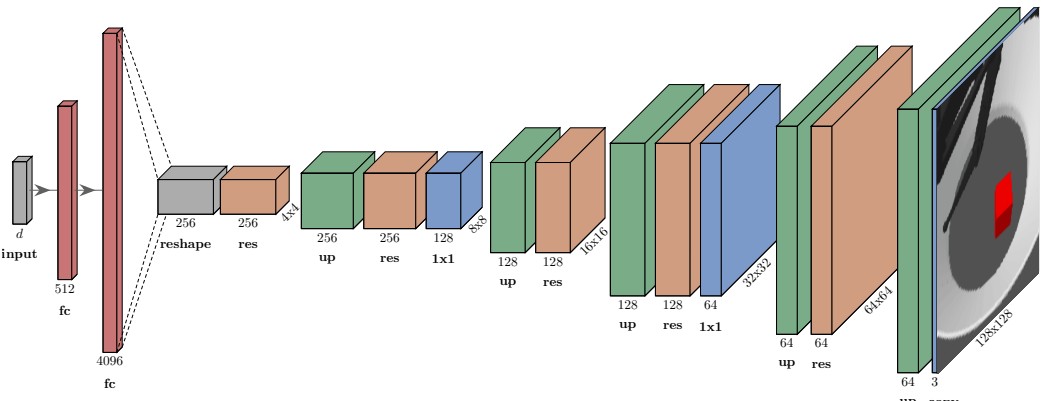

Figure 8: Schemes of the encoder (top) and decoder (bottom) architectures. In both schemes, information flows left to right. Blue blocks represent convolutional layers: those labeled "conv" have 5x5 kernels and stride 2, while those labeled "1x1" have 1x1 kernels. Each orange block represents a pair of residual blocks (implementation details of a residual block are provided in Table 3). Green blocks in the encoder represent average pooling with stride 2, and those in the decoder denote bilinear upsampling by a factor of 2. Red blocks represent fully-connected layers. The block labeled "norm" indicates layer normalization. Dashed lines denote tensor reshaping.

# B ADDITIONAL RESULTS

## B.1 DATASET CORRELATIONS

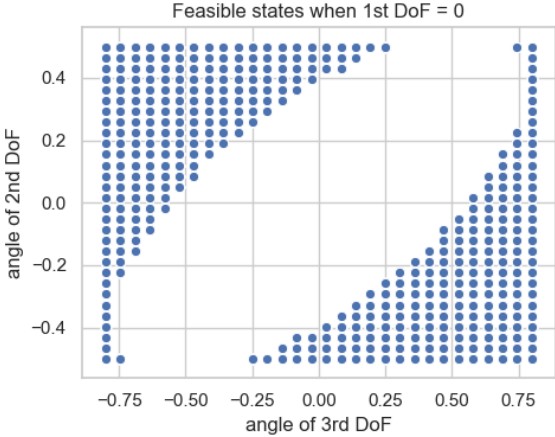

Figure 9: Feasible states of the 2nd and 3rd DoF when the angle of the 1st DoF is 0. Angles are in radians.

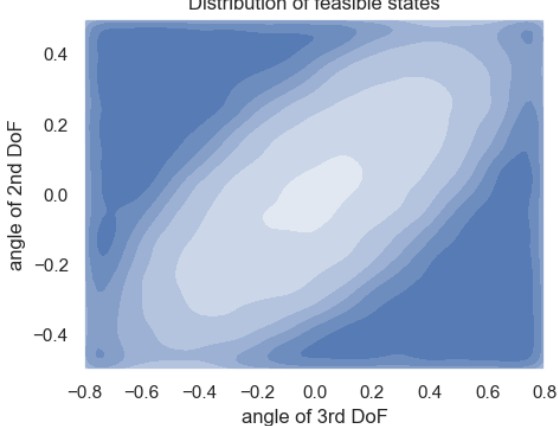

Figure 10: Density of feasible states of 2nd and 3rd DoF over the whole training dataset. Darker shades of blue indicate regions of higher density. Angles are in radians.

### B.2 Samples and Reconstructions

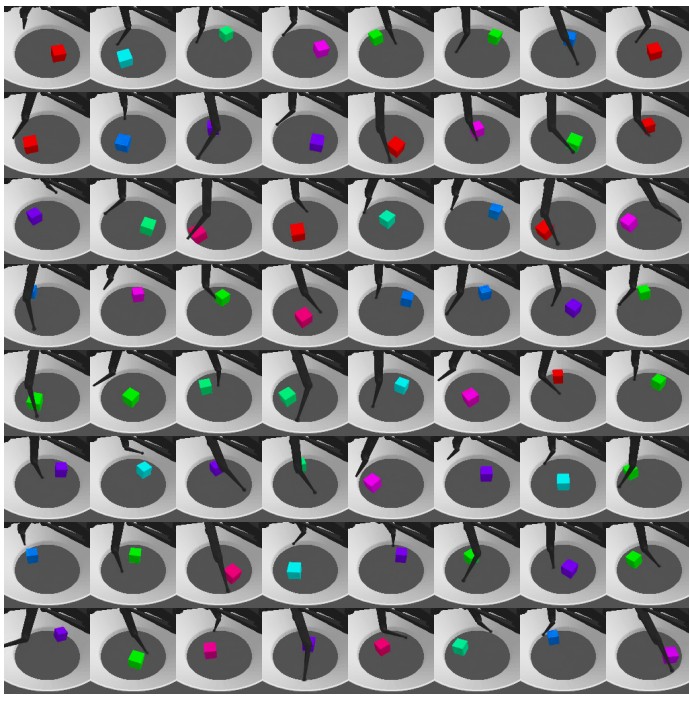

Figure 11: Samples generated by a trained model. This model was selected based on the ELBO.

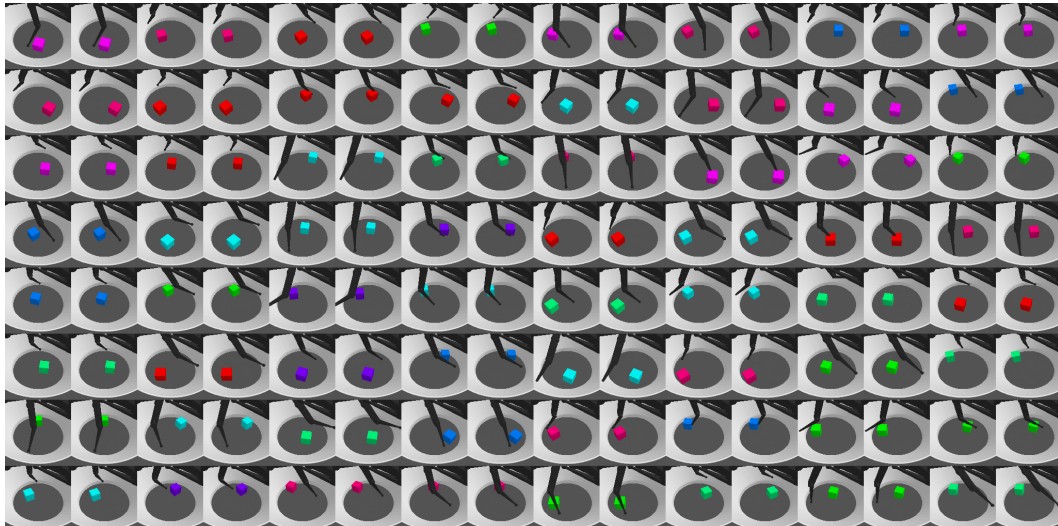

Figure 12: Input reconstructions by a trained model. This model was selected based on the ELBO. Image inputs are on odd columns, reconstructions on even columns.

### B.3 LATENT TRAVERSALS

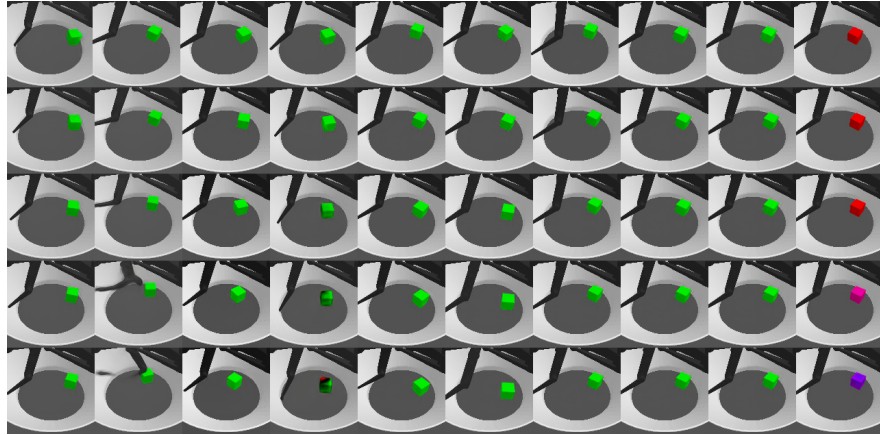

(a) Low disentanglement

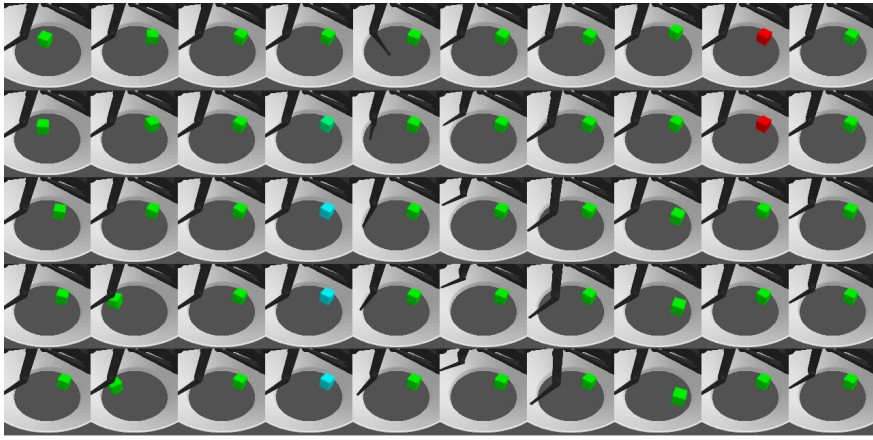

(b) Medium disentanglement

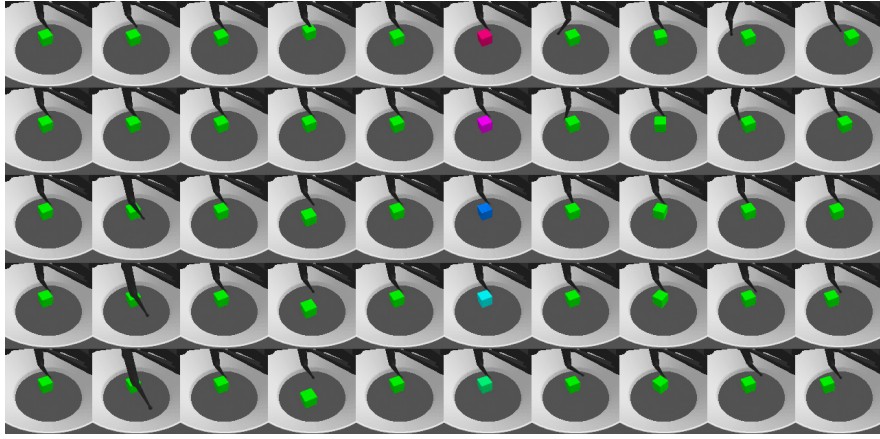

(c) High disentanglement

Figure 13: Latent traversals for a model with low DCI score (0.15) in (a), medium DCI score (0.5) in (b), and high DCI score (1.0) in (c).

### B.4 UNSUPERVISED METRICS AND DISENTANGLEMENT

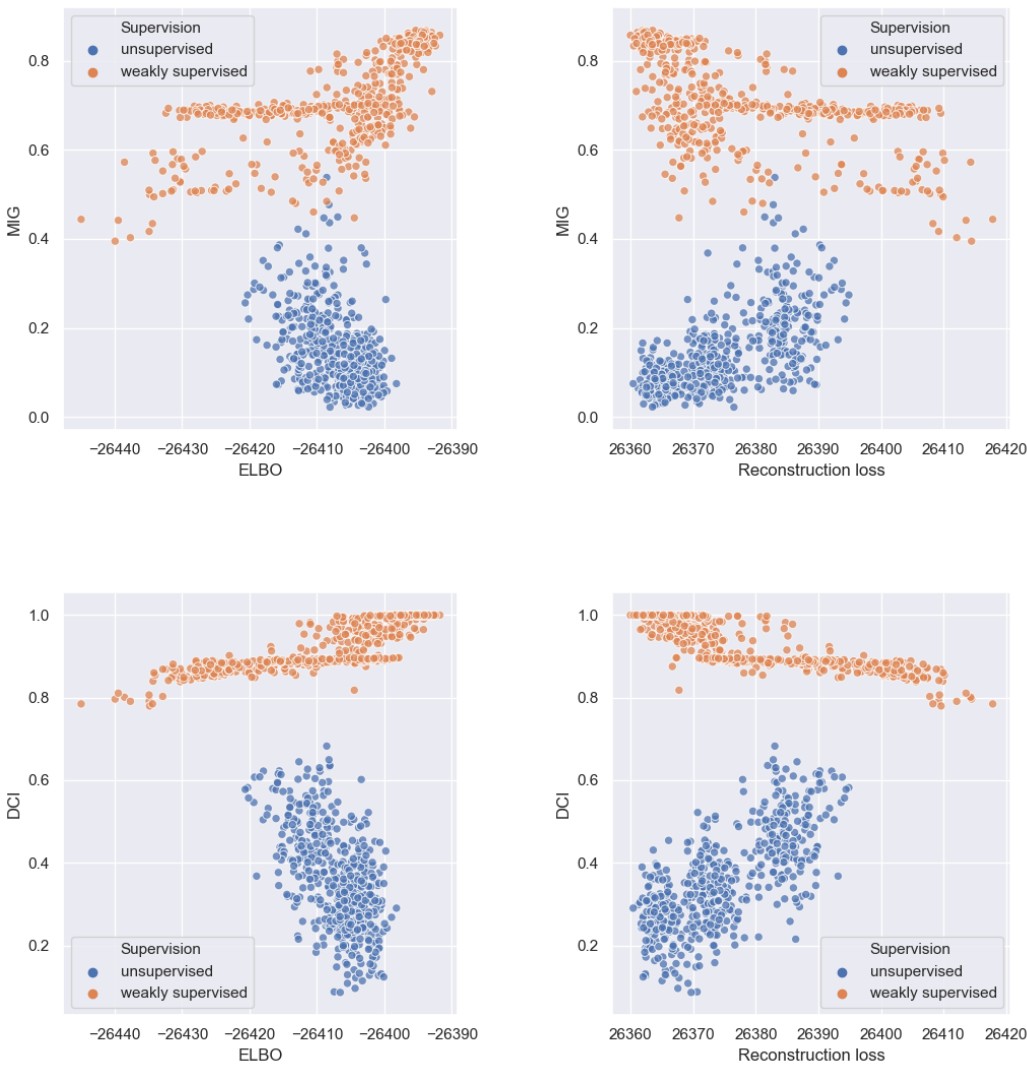

Figure 14: Scatter plots of unsupervised metrics (left: ELBO; right: reconstruction loss) vs disentanglement (top: MIG; bottom: DCI) for 1,080 trained models, color-coded according to supervision. Each point represents a trained model.

## B.5 Out-of-Distribution Transfer

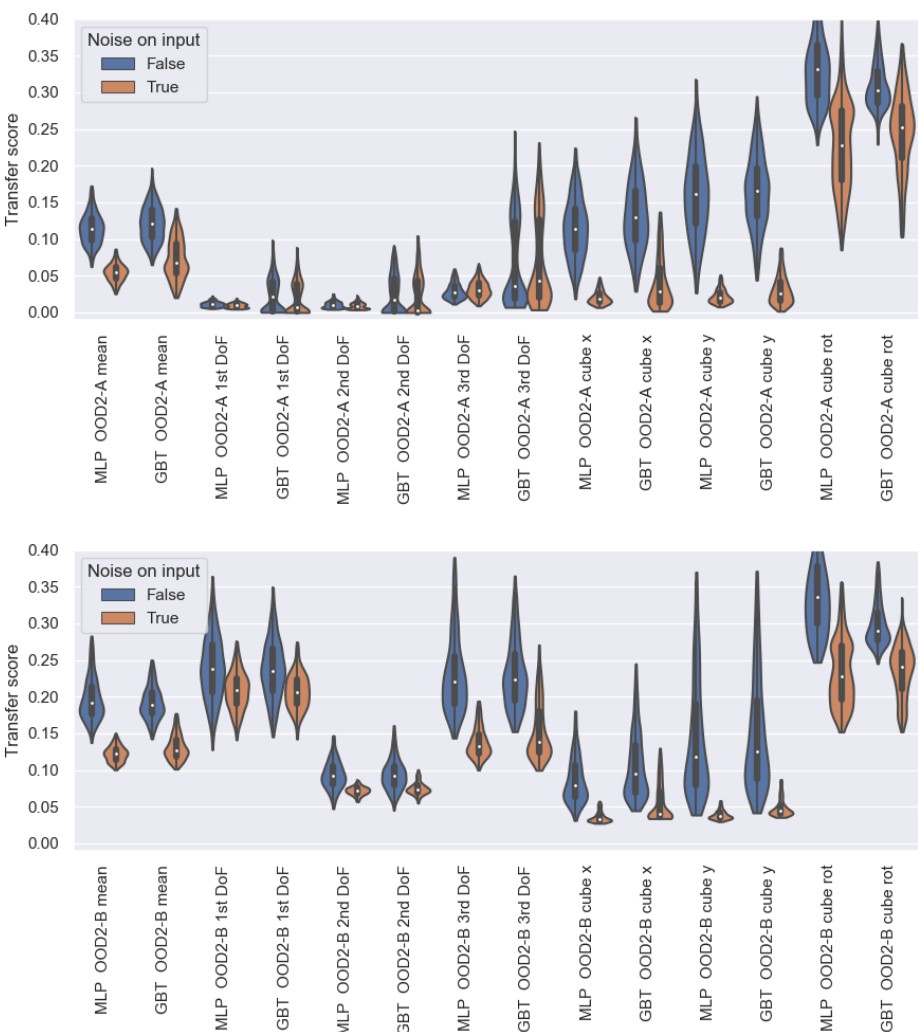

Figure 15: Transfer metric in OOD2-A (top) and OOD2-B (bottom) settings, decomposed according to the factor of variation and presence of input noise. When noise is added to the input during training, the inferred cube position error is relatively low (the scores are the mean absolute error, and they are normalized to $[0, 1]$). This is particularly useful in the OOD2-B setting (real world) where the joint state is anyway considered known, while object position has to be inferred with tracking methods.

## B.6 OUT-OF-DISTRIBUTION RECONSTRUCTIONS

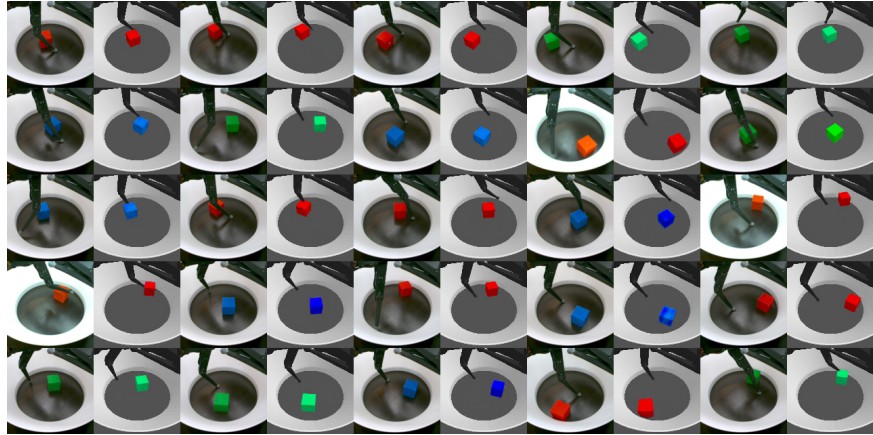

(a) Low disentanglement

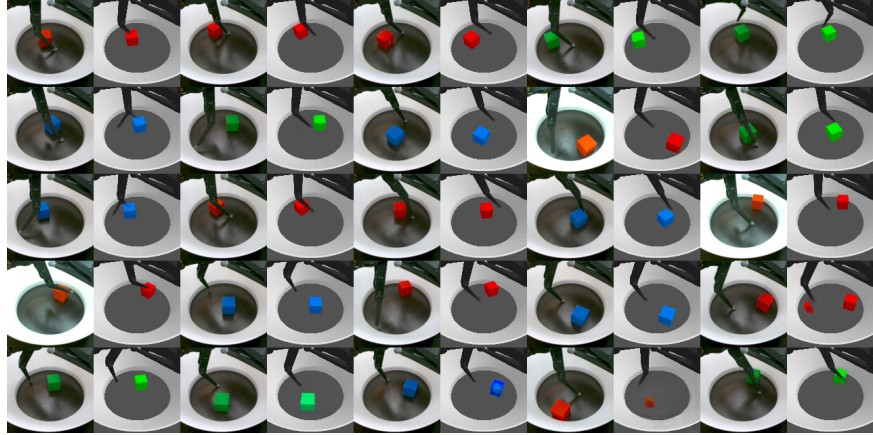

(b) Medium disentanglement

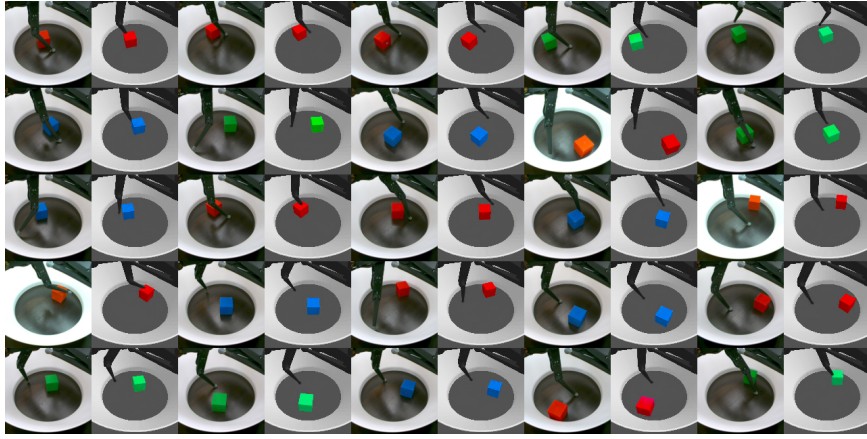

(c) High disentanglement

Figure 16: Reconstructions of real-world images (OOD2-B) for a model with low DCI score (0.15) in (a), medium DCI score (0.5) in (b), and high DCI score (1.0) in (c).

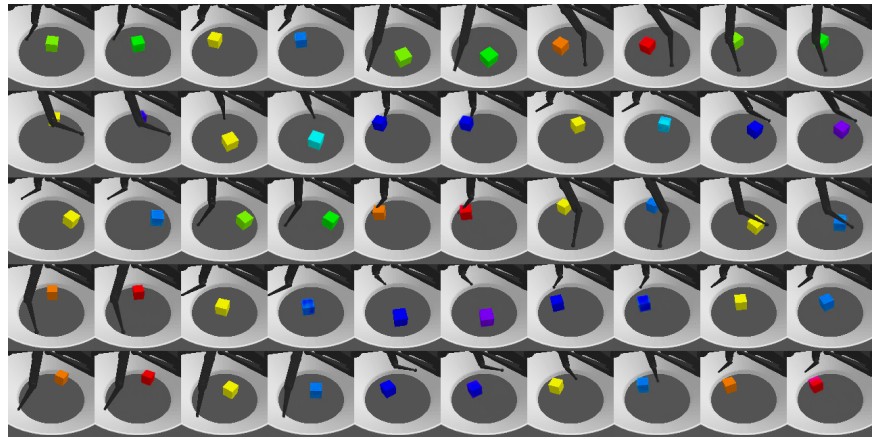

(a) Low disentanglement

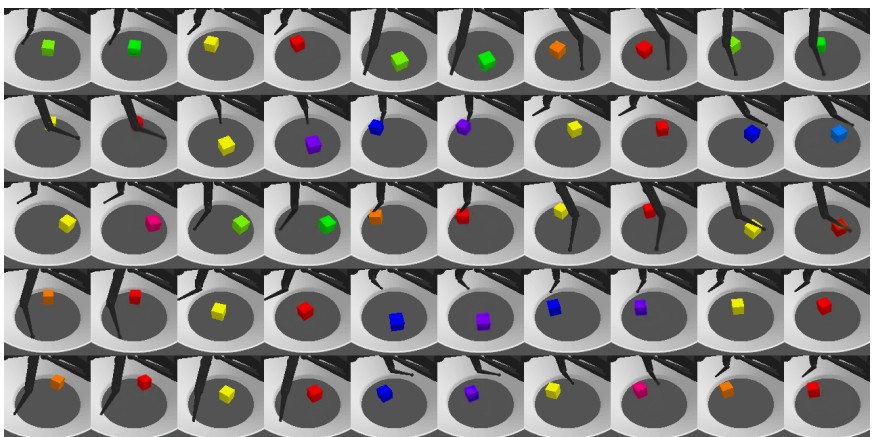

(b) Medium disentanglement

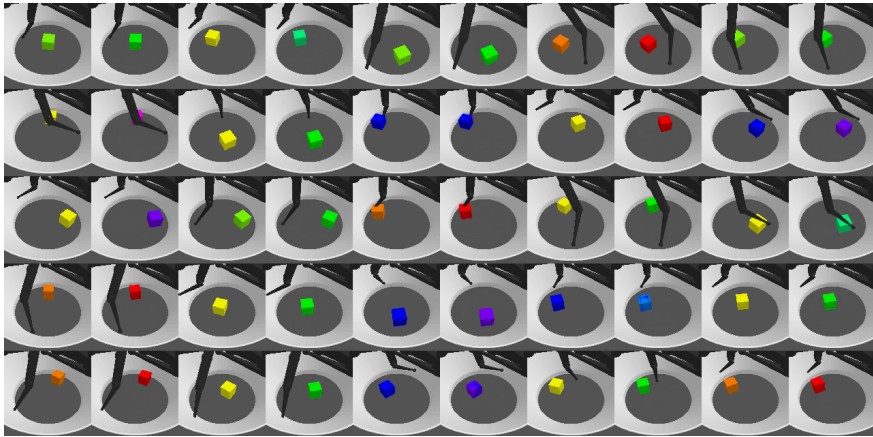

(c) High disentanglement

Figure 17: Reconstructions of simulated images with encoder out-of-distribution colors (OOD2-A) for a model with low DCI score (0.15) in (a), medium DCI score (0.5) in (b), and high DCI score (1.0) in (c).

