# OpenReview forum: "On the Transfer of Disentangled Representations in Realistic Settings"
_ICLR.cc/2021/Conference — ICLR 2021 Poster_

### Official Review · AnonReviewer4 · 2020-10-26
**The new standard dataset for complex disentanglement learning**

**Rating:** 9
**Confidence:** 4

**Review:**

## Summary

The paper presents a new, more complex, dataset for the use of disentangled representation learning. The dataset is based on real and simulated images of the trifinger robot platform. There are 7 factors of variation with high-resolution measurements of these factors. The dataset contains over 1 million simulated images and another ~1000 annotated images of a real trifinger robot arm.

The authors also present a new neural architecture to scale disentanglement on more complex datasets and present a large empirical study on the performance of various techniques on out-of-distribution downstream task performance.


## Quality & Clarity
The paper itself is well written, with a structure that effectively guides the reader through the work and results.

The dataset, significance and experiments are clearly outlined.

## Originally & Significance

The novelty of the dataset is clear. Providing a complex disentanglement dataset where the underlying factors of variation are inherently correlated. The in- and out-of-distribution experiments are possible because of the presence of both synthetic and real-world data.

The experiments run are repeated multiple times and the results are convincing. They use both unsupervised and weakly supervised approaches and the results are both intuitive and supported by the literature.

The experiments on out-of-distribution representation transfer are interesting and show that disentangled representations can lead to better transfer to out-of-distribution tasks.


## Outcome Rationale

This dataset is likely to be extremely useful to the community going forward and work disentangled representation learning is likely to benefit from it. The experimental setups are sensible and the largescale benchmarks support the use of disentangled representations when transferring from simulated to real-world scenarios.

---

> ### Author Response · Authors · 2020-11-17
> **Response to Reviewer 4**
>
> We thank the reviewer for taking the time and effort to review our paper. It is encouraging to see the reviewer considers our work convincing and important, and in particular the dataset and the experimental setup to be potentially extremely useful to the community going forward.

---

### Official Review · AnonReviewer2 · 2020-10-27

**Rating:** 7
**Confidence:** 5

**Review:**

The work has sufficient quality, and is sufficiently clear and original. For detail, please see the below pros and cons.

Overall Pros:
- Dataset contains dependencies in FoV, not artificially induced but present due to the attempt at realism in the simulation, a realistic aspect which hasn't been addressed in previous work.
-The complete set of factors (e.g. product of number of possible values for each factor), totals to 2.92B, thus the dataset contribution allows for addressing the generalization problem which hasn't been addressed in previous work.
-Disentanglement metrics notably cannot be evaluated on acquired camera images, so the recording of an annotated dataset under the same conditions in a real-world setup enables researchers to study this problem in a more controlled manner.
-The observation that deeper and wider autoencoders can arguably scale to datasets with fine-grained factors is an informative observation for the community.
-Similar to [1], the results collected in the extensive experimental study provide information to the community which can be analyzed and built upon in future work.
-The OOD evaluation is novel, interesting, and informative. Though the downstream task is quite connected with the disentanglement evaluation itself, the consideration of this setting is a contribution to the community.
-The negative observation in sim-to-real transfer is a net positive for this work, as it is informative to the community in specifying a problem which can be worked on in future research.
-The observation that adding input noise in training, a common tool in classic sim-to-real robotics work, is useful for transfer in representations is practical and meaningful.

Overall Cons:
- For the evaluation prior to OOD evaluation, it appears disentanglement is evaluated on the full collected dataset. If that is the case, though the dataset itself provides the ability to evaluate generalization, the evaluation prior to OOD evaluation does not evaluate generalization. It is not clear then if overfitting played a role in the observed results.
- Fixing k=1 for the weakly supervised method implies control over the data generating process often unavailable in practical applications. At the same time, said experiment can be extended to test how performance degrades with a mismatch in assumptions. Considering k != 1, as well as PCL [2] and SlowVAE [3] (which assume laplacian transitions) could provide interesting comparisons.
- The authors state that "many" trained models fully disentangle the factors, but this is "only" possible in the weakly supervised scenario. Clarifying details would be beneficial here, is it true that none of the unsupervised models fully disentangled the factors? If some did, was there an underlying correlation between successful models that can be discussed, or was it simply random? Is using latent traversals alone as a basis sufficient for claiming "full disentanglement"? More specific statements in the description would be desirable.
- The authors appear to pre-assume that since latent traversals agree with the theory of [1], quantitative metrics which do not score weakly supervised models over unsupervised models are "ineffective at capturing disentanglement in this setting". This reads as confirmation bias and should clarified, either by justifying why metrics which do not agree with latent traversals are "ineffective", or presenting the information without implying judgement on whether the metrics are "ineffective" or not.
- "We stress that the OOD2 scenario, which is typically not studied extensively, is only possible because the representations are not trained on the entire dataset." Does this mean the representations learned for OOD evaluation are trained on less data than what was considered for disentanglement evaluation? Clarifying details would be beneficial here, as this standalone statement simply yields unanswered questions.
- How was D1 selected from D, by what sampling process, and how was the number of samples decided upon? For in-distribution generalization, how was the split between training and held-out set in D1 conducted, what was the percentage for the split, and how was it selected? Clarifying detail would be beneficial.
- "Since the values are normalized, we can take the average of the MAE over all factors (except for the FoV which is OOD)." Why can you not take the average of the MAE over the FoV which are OOD? This statement could be interpreted as some factors being OOD while some not, when a split within the factor set was not mentioned previously. This sentence requires clarification.
- Many questions brought up by Section 4 (such as the above), are addressed in part in the experimental setup section for Section 5. I'd suggest providing said information earlier so the confusion yielded in Section 4 is mitigated.
- Notably, the color hues simply mentioned in Table 1, without any explanation on why they are not included within Table 1, appear to be what is ablated upon for OOD1. It would have been beneficial to have earlier clarification on this point. Furthermore, distribution shift in the color hue but all underlying factors being the same is specific type of distribution shift not clarified to the reader. Consideration of discrepancies in renderer would be interesting, but more importantly, discrepancies in the factor value set. The distribution shift considered here is shift in nuisance factors, a useful test but limited in its scope.
- "Our results therefore suggest that highly disentangled representations are useful for generalising out-of-distribution as long as the encoder remains in-distribution" This statement can be seen as a bit misleading given the data is only out-of-distribution in terms of nuisance factors, not the factors the model was evaluated on disentanglement with respect to. It's intuitive that a highly disentangled representation will capture the ground truth factors well, and thus be more robust to shift in nuisance factors, and it is useful to show this, but the fact that we are looking at shift in nuisance factors should be clear to the reader.
- The justification for training half of the models with gaussian noise is provided in the final page of the paper. As with the previous comments, it would be beneficial to the reader to make this clear when introducing the experimental design decisions.

Conclusion: This work provides significant contribution in experimental analysis on the performance of disentanglement with the inclusion of realistic complexities. Both the positive and negative observations are clear, interesting, and informative. While I suggested many ablations which could provide the reader with further information, I don't see the lack of inclusion of said ablations as an issue in the work, the experimental study is already extensive. The main issue I see with the work as it is merely comes from a writing standpoint, ensuring the statements made are justified, and ensuring the detail is there in the descriptions for the reader to understand, and most importantly, be able to reproduce, said results.

Overall, I see this is a valuable contribution, but would like to see improvement in the writing given the points brought up in the revision.

Questions:
- Can you describe how given a particular state for the factors of variation, how said factors are processed to render the observation? A description of how each factor individually influences the observation would be beneficial in the main paper, if only briefly, such that the reader can see why it is (1) challenging and (2) requires modeling of fine details.
- "The training set for the VAEs contains 8 randomly chosen color hues." How do these color hues differ from the cube color hues? How do they specifically affect the rendering? Clarifying detail for this statement is needed.
- Were any ablation studies performed for the model architecture? Many model details are presented without any experimental testing discussed, an ablation study would be informative to the reader.
- Did you test the behavior of models with latent_dim < factor_dim, e.g. latent space dimensionality of 5, as well as latent_dim = factor_dim, e.g. latent space dimensionality of 7? Such results could be interesting to see if, and how much, performance degrades not only as latent_dim progressively increases from factor_dim, but also when latent_dim progressively decreases from factor_dim, considering the equality case as well.
- " Finally, note that the BetaVAE and FactorVAE metrics are not straightforward to be evaluated on datasets
that do not contain all possible combinations of factor values." Could you detail why they are not straightforward to evaluate? Clarifying detail would be helpful.
- Did the authors consider other unsupervised model selection techniques, such as UDR, to see if this under or outperformed the weakly supervised loss selection method?
- Why the choice of MAE? Was this choice ablated on?

[1]: Francesco Locatello, Stefan Bauer, Mario Lucic, Sylvain Gelly, Bernhard Scholkopf, and Olivier ¨
Bachem. Challenging common assumptions in the unsupervised learning of disentangled representations. In International Conference on Machine Learning, 2019.

[2]: Aapo Hyvärinen and Hiroshi Morioka. Nonlinear ica of temporally dependent stationary sources. In Proceedings
of Machine Learning Research, 2017.

[3]: David Klindt, Lukas Schott, Yash Sharma, Ivan Ustyuzhaninov, Wieland Brendel, Matthias Bethge,
and Dylan Paiton. Towards nonlinear disentanglement in natural data with temporal sparse coding.
arXiv preprint arXiv:2007.10930, 2020.

---

> ### Author Response · Authors · 2020-11-17
> **Response to Reviewer 2**
>
> We are grateful to the reviewer for the very extensive and constructive feedback, which already helped us improve the clarity of the paper. We are pleased to read that the reviewer considers our work “a significant contribution in experimental analysis on the performance of disentanglement with the inclusion of realistic complexities”. We revised the manuscript, especially on the OOD evaluation setup in Section 4 and Appendix A, to address issues pointed out by the reviewer in cons 3, 5, 6, 7, 8, 9, 10, and question 2. We address the remaining reviewer’s concerns point-by-point below.
>
> Cons:
>
> - 1\) We chose this approach for consistency with previous work [1,2,3]. However, we agree that it would be interesting to evaluate disentanglement metrics on held-out examples.
>
> - 2\) Comparing the effectiveness of different disentanglement methods in our dataset is indeed interesting and relevant for the community. However, since evaluating the relationship between disentanglement and OOD generalization already required significant compute, we chose to limit the study in terms of disentanglement methods.
>
> - 3a\) Many of the weakly supervised models are highly disentangled, while none of the unsupervised models are. This can be seen from: (1) the main text, where we report how many weakly supervised models have >99% DCI score, (2) the violin plots in Fig. 3 left, and (3) the scatter plots in Appendix B.4 in the new version of the paper.
>
> - 3b\) Regarding latent traversals: In our experience, disentanglement metrics do not always behave as expected [4]. Model visualizations such as latent space traversals remain the gold standard for assessing disentanglement.
>
> - 4\) In our study we observed from visual inspection (see point 3b) that SAP and Modularity give similar scores to entangled and disentangled models, which is not desirable, while DCI and MIG more reliably differentiate between them. Limitations of disentanglement metrics are also discussed in [4].
>
> - 5\) No, we evaluate the same representations on disentanglement and OOD generalization. Thanks for pointing this out - we removed that sentence as it was misleading.
>
> - 6\) D1 was selected from D by sampling images where the cube color is in the desired set. Regarding in-distribution generalization: we only report it in terms of the GBT10000 metric, for consistency with previous work. Both GBT10000 and the downstream predictors are trained on 10k samples and tested on 5k as in [1,2,3].
>
> - 7-10\) We clarified the setup of the OOD evaluation in the revised manuscript. About 10): note that the test sets are OOD in terms of cube color, which is also taken into account when evaluating disentanglement. So it is not true that the data is not out-of-distribution in terms of the factors the model was evaluated on disentanglement with respect to.
>
> - 11\) The justification for Gaussian noise is already given in the experimental setup paragraph in Section 3.
>
>
>
> Questions:
>
> - 1\) We used the bullet physics engine to render images of this scene given specific properties, i.e. the factors of variation. We assumed that the factor names were self-explanatory, but we are happy to include more detailed information if deemed helpful. Note that the separate factors are also visible in the latent traversals from the models with high disentanglement.
>
> - 2\) The 8 color hues are the cube color hues. We realize this might be unclear and fixed it in the revised version.
>
> - 3\) We performed basic ablation studies to find an architecture that would allow learning fully disentangled representations. These are only briefly mentioned in the appendix, without numerical results.
>
> - 4\) We sweeped only through latent dimensions bigger than the number of ground truth factors of variation, as is typically done in other disentanglement studies [1,2,3].
>
> - 5\) It requires access to images with arbitrary combinations of factors of variation, which is not possible in our dataset, as it is not generated on-the-fly.
>
> - 6\) This is a very interesting idea which we left for future work. Note that the correlation we observed between ELBO (or reconstruction loss) and disentanglement is very strong - in fact, it is stronger than the one reported in the UDR paper for the unsupervised setting.
>
> - 7\) Any measure of mean error would work, as we are mainly interested in the relative performance of different models. We chose the MAE for its relatively intuitive interpretation.
>
>
>
> [1] Locatello et al. "Challenging common assumptions in the unsupervised learning of disentangled representations." ICML 2019.
>
> [2] Locatello et al. "On the fairness of disentangled representations." NeurIPS 2019.
>
> [3] van Steenkiste et al. "Are Disentangled Representations Helpful for Abstract Visual Reasoning?" NeurIPS 2019.
>
> [4] Locatello et al. “A Sober Look at the Unsupervised Learning of Disentangled Representations and their Evaluation.” JMLR, 2020.

---

### Official Review · AnonReviewer3 · 2020-10-27
**Not technically sound and lack of sufficient evaluation**

**Rating:** 2
**Confidence:** 5

**Review:**

The authors proposed a unique learning scheme for representation disentanglement. However, unlike infoGAN or ACGAN which explicitly learn disjoint feature representations for describing the attributes of interest (via unsupervised and supervised settings, respectively), the authors chose to address this task in a questionable "weakly supervised setting". More specifically, the authors chose to train AE-like model using pairwise images, which the difference between each pair of the inputs is only associated with one attribute of interest (e.g., angle, position, etc.).

For some reasons, the authors expected such a training scheme would result in learning feature representation in which only "one" feature dimension would reflect such attribute differences. This is a very strong assumption, since it is very likely that more than one feature dimensions would correspond to such changes.

Moreover, assuming that precisely one feature dimension would be associated with the attribute of interest by feeding in a pair of images with exactly this attribute change would not be practical either. Most real-world images would be complex and contain multiple attributes. Making this assumption would imply that the training images are not realistic.

As for the evaluation, there is no comparison to any baseline or SOTA representation disentanglement methods, I found the quantitative metrics selected by the authors not sufficiently informative or supportive either. Most importantly, the authors claimed that the features trained by VAE allowed improved performances (e.g., Figs 3~5). Since VAE are simply trained in a unsupervised way (even the authors called their setting a weakly supervised setting), I see no evidence why the resulting features would be any different from those derived from standard VAEs, and why improved disentanglement results could be achieved.

Based on the above observations and remarks, I feel that the authors would not be able to deliver a work which is technically strong with sufficiently complete evaluation. Therefore, I do not think this paper is above the ICLR standard for acceptance.

---

> ### Author Response · Authors · 2020-11-17
> **Response to Reviewer 3**
>
> We thank the reviewer for the feedback. We believe most of this reviewer’s concerns to be due to misunderstandings of the primary objectives and contributions of this work, which we hope to clarify below.
>
> 1\) On the weakly supervised setting:
>
> 1a\) “The authors proposed a unique learning scheme [...]”: In fact, we only use existing SOTA methods for disentangled representation learning: we train beta-VAEs (1) without supervision by optimizing the ELBO, and (2) with weak supervision using Ada-GVAE [1]. The contributions of this work do not include proposing a new method for disentangled representation learning.
>
> 1b\) “the authors expected such a training scheme [...]”: This training scheme had already been shown to be more effective than unsupervised learning [1]. Our results confirm this.
>
> 1c\) “Moreover, assuming [...] would not be practical”: We chose k=1 because this has been shown to lead to higher disentanglement than k>1 [1]. We stress that we are not interested in the specific learning method, but rather in evaluating the role of disentanglement. The question of whether Ada-GVAE with k=1 is generally practical is certainly relevant, but orthogonal to this paper (see [1] for a discussion).
>
> 2\) Comparison to baseline or SOTA representation disentanglement methods.
> The focus of our study is on the effect of disentanglement on OOD generalization - not on comparing different disentangled representation learning methods. We employ SOTA disentanglement learning methods (from disentanglement_lib [2]) to learn representations with a wide range of disentanglement, which in turns allows a sound empirical study on downstream OOD generalization.
>
> 3\) Quantitative metrics selected by the authors not sufficiently informative or supportive.
> The selected disentanglement metrics are widely used in the context of autoencoder-based representation learning methods [2-9]. Nevertheless, we gladly welcome any suggestions for more informative disentanglement metrics.
>
> 4\) “Since VAE are simply trained in a unsupervised way [...] I see no evidence why the resulting features would be any different from those derived from standard VAEs, and why improved disentanglement results could be achieved.”
> We train VAEs either with the standard unsupervised approach or with the Ada-GVAE weakly supervised method. We want to stress that Ada-GVAE is a fundamentally different training algorithm that relies on weak labels, so it is not true that all our VAEs are trained in an unsupervised way. Moreover, we do not claim the weakly supervised representations should necessarily be more disentangled - we simply observe empirically that they tend to be, which is also in agreement with results in [1].
>
> [1] Locatello et al. “Weakly-supervised disentanglement without compromises.” ICML 2020.
>
> [2] Locatello et al. "Challenging common assumptions in the unsupervised learning of disentangled representations." ICML 2019.
>
> [3] Chen et al. “Isolating sources of disentanglement in variational autoencoders.” NeurIPS 2018.
>
> [4] Ridgeway and Mozer “Learning deep disentangled embeddings with the f-statistic loss.” NeurIPS 2018.
>
> [5] Eastwood and Williams “A framework for the quantitative evaluation of disentangled representations.” ICLR 2018.
>
> [6] Kumar et al. “Variational inference of disentangled latent concepts from unlabeled observations.” ICLR 2018.
>
> [7] Locatello et al. "On the fairness of disentangled representations." NeurIPS 2019.
>
> [8] Locatello et al. “A Sober Look at the Unsupervised Learning of Disentangled Representations and their Evaluation.” JMLR, 2020.
>
> [9] van Steenkiste et al. "Are Disentangled Representations Helpful for Abstract Visual Reasoning?" NeurIPS 2019.

---

### Official Review · AnonReviewer1 · 2020-10-27

**Rating:** 5
**Confidence:** 4

**Review:**

Summary:
This paper identifies that traditional datasets used for learning disentangled representation have several shortcomings such as no correlation between variables and simple structure.
It proposes a new dataset that has 1M higher-resolution simulated images along with 1K annotated real-world images of the same setup and gives analysis on disentangled representations on the dataset.
Its results suggest that disentangled representations can result in better out of distribution task performances.

==================================

Strength:
- Identified weaknesses of the previous datasets and proposed a new dataset that exhibits correlations between different variables. This is an important aspect of real-world scenarios.
- Provided thorough experiments on disentangled representations and their metrics on the proposed dataset.

Weakeness:
- Experimental results for showing more disentangled representation results in better OOD task performances is somewhat expected. Unlike the title, it is not clear if the paper shows sufficient transfer of disentangled representations in realistic settings. It would be great to propose a way to make the generalization better for the settings the models have trouble with (OOD2 generalization).
- This paper tried one approach by adding noise during training, but it leads to my second concern that the real world observation is very similar to the simulated data. With some gaussian noise, they would look similar. Therefore, it might not be sufficient to show that we can use this approach for sim-to-real transfer.
- The paper claims that the proposed dataset (which is interesting) is challenging and highly complex, but the rendered images look easy enough for a simple neural network model to learn to reconstruct. Datasets like CLEVR, although they might not be used for measuring disentanglement, seems more complex and exhibits occlusions.

==================================

While the proposed dataset is interesting, I am not confident this paper is showing evidence for the usefulness of disentangled representations for transfer learning in realistic settings. It is also not clear if the dataset is more useful than others because of the reasons stated above.

---

> ### Author Response · Authors · 2020-11-17
> **Response to Reviewer 1**
>
> We thank the reviewer for the helpful feedback and appreciate the comments that we offer a solution to weaknesses of previous datasets and carry out thorough experiments. We have updated the paper and we hope some of the issues might be resolved. We address the reviewer’s concerns individually below.
>
> 1a\) On the context, scope and contributions of this work.
> We would like to emphasize that the primary goal of our work was to empirically investigate transfer capabilities under different OOD scenarios of disentangled representations using popular disentanglement learners beyond synthetic toy datasets. We called our setting “realistic” because (1) it exhibits challenges of a real-world robotic setup, with real images as well as simulated images generated with the bullet physics engine renderer, (2) we study transfer on the real setup (using the accompanied annotated images) and (3) it opens up the possibility for evaluations on downstream-tasks such as RL that can be deployed and tested on the real robot. Nevertheless, we would greatly appreciate any suggestions on how to improve the title.
>
> 1b\) On the assumption that disentanglement helps for transfer.
> Although one might expect that disentanglement helps for OOD generalization, this is actually not obvious. Therefore, we believe there is value in exploring this empirically, especially since we are not aware of literature that investigates this in a large-scale study. Interestingly, our results indicate for example that disentanglement is not necessarily useful in the OOD2 setting, which provides further proof that these relationships should be thoroughly investigated.
>
> 1c\) On novel methods to improve OOD2 generalization.
> We believe our results offer insights that may help progress towards achieving broader out-of-distribution generalization, which is a long-standing challenge in machine learning. Possible research avenues for improving generalization in the OOD2 scenario may include stronger inductive biases and different imposed structures on the latent space. However, we leave exploring these approaches to future work.
>
> 2\) On sim-to-real results with Gaussian noise.
> In our study we observed that adding noise seems to be more beneficial for generalization than disentanglement, and we believe this to be an important insight. However, we do not claim that this would be sufficient for effective sim-to-real transfer, which is in fact not the main focus of our work.
>
> 3\) The dataset is not really challenging.
> We want to stress that in the context of disentanglement studies this dataset is certainly more challenging and complex than previous ones. Even though it might not seem so hard for neural networks, previous work on disentanglement only focused on very simple models (encoder and decoder have 4 conv layers and 2 FC). Moreover, previous architectures are often just not enough for this type of data (see for example reconstructions on simpler datasets like SmallNORB in [1]).
> We agree CLEVR is a very interesting dataset, but as the reviewer points out it’s not straightforward how disentanglement should be interpreted in this context. About occlusions, note that our dataset also contains heavy occlusions, unlike previous disentanglement datasets. Moreover, our dataset is derived from a robotic platform so one can directly move towards RL tasks in future work, unlike with CLEVR.
>
> [1] Locatello et al. "Challenging common assumptions in the unsupervised learning of disentangled representations." ICML 2019.

---

### Decision · Program_Chairs · 2021-01-07
**Final Decision**

**Decision:**

Accept (Poster)

**Comment:**

This paper introduces a new dataset for evaluating disentanglement and its impact on out of distribution generalization based on the trifinger robotics platform. Using this dataset, the authors rigorously investigate the performance of beta-VAEs in this setting under a number of conditions, finding that weak supervision is necessary to induce disentangled representations, and that, perhaps surprisingly, disentanglement does not help for sim2real settings despite the similarity between the simulator and the real data. Reviewers were divided on the work, but had a number of concerns related to the claims of novel architecture, comparisons to baselines, and issues with the clarity of the paper, some of which were addressed in the authors' response. I agree with some of these concerns, particularly with respect to the claims of novel architectures since the modifications could simply be viewed as tweaking hyperparameters and are not rigorously compared to baselines. However, I think the novelty of the dataset and the rigorous evaluation of OOD generalization settings is likely to be valuable enough to the community to merit acceptance. I'd encourage the authors, however, to tone down some of the claims regarding the architecture (or provide sufficient baseline comparisons), and instead focus on the dataset and the OOD results. I recommend acceptance.